



# Reviews and syntheses: Greenhouse gas exchange data from drained organic forest soils – a review of current approaches and recommendations for future research

Jyrki Jauhiainen[1,2], Jukka Alm[3], Brynhildur Bjarnadottir[4], Ingeborg Callesen[5], Jesper R. Christiansen[5], Nicholas Clarke[6], Lise Dalsgaard[7], Hongxing He[8], Sabine Jordan[9], Vaiva Kazanavičiūtė[10], Leif Klemedtsson[11], Ari Lauren[3], Andis Lazdins[12], Aleksi Lehtonen[1], Annalea Lohila[13,14], Ainars Lupikis[12], Ülo Mander[15], Kari Minkkinen[2], Åsa Kasimir[11], Mats Olsson[9], Paavo Ojanen[2], Hlynur Óskarsson[16], Bjarni D. Sigurdsson[16], Gunnhild Søgaard[7], Kaido Soosaar[15], Lars Vesterdal[5], and Raija Laiho[1]

[1] Natural Resources Institute Finland (Luke), Box 2, FI–00791 Helsinki, Finland

[2] Department of Forest Sciences, University of Helsinki, Box 27, FI–00014, Helsinki, Finland

[3] Natural Resources Institute Finland (Luke), FI–80100 Joensuu, Finland

[4] Department of Education, University of Akureyri, IS-600 Akureyri, Iceland

[5] Department of Geosciences and Natural Resource Management, University of Copenhagen, DK–1958 Frederiksberg C, Denmark

[6] Department of Terrestrial Ecology, Norwegian Institute of Bioeconomy Research (NIBIO), Box 115, N–1431 Ås, Norway

[7] Department of Forest and Climate, Norwegian Institute of Bioeconomy Research (NIBIO), Box 115, N–1431 Ås, Norway

[8] Department of Biological and Environmental Sciences, University of Gothenburg, Box 461, SE–40530 Gothenburg, Sweden

[9] Department of Soil and Environment, Swedish University of Agricultural Sciences, Box 7014, SE–75007 Uppsala, Sweden

[10] Lithuanian State Forest Service, LT-51327 Kaunas, Lithuania

[11] Department of Earth Sciences, University of Gothenburg, Box 460, SE–40530 Gothenburg, Sweden

[12] Latvian State Forest Research Institute (Silava), Salaspils, LV–2169, Latvia

[13] INAR Institute for Atmospheric and Earth System Research/Physics, Faculty of Science, University of Helsinki, Box 68, FI-00014, Helsinki, Finland

[14] Finnish Meteorological Institute, Climate System Research, Box 503, FI–00101 Helsinki, Finland

[15] Department of Geography, University of Tartu, EE-51014 Tartu, Estonia

[16] Agricultural University of Iceland, IS-311 Hvanneyri, Borgarnes, Iceland

*Correspondence to*: Jyrki Jauhiainen (jyrki.jauhiainen@helsinki.fi)

*Keywords:* greenhouse gases, emission factors, peatlands, organic soils



**Abstract.** Drained organic forest soils in boreal and temperate climate zones are believed to be significant sources of the greenhouse gases (GHG) carbon dioxide ($CO_2$), methane ($CH_4$) and nitrous oxide ($N_2O$), but the annual fluxes are still

highly uncertain. Drained organic soils exemplify systems where many studies are still carried out with relatively small resources, several methodologies and manually operated systems, which further involve different options for the detailed design of the measurement and data analysis protocols for deriving the annual flux. It would be beneficial to set certain guidelines for how to measure and report the data, so that data from individual studies could also be used in synthesis work based on data collation and modelling. Such synthesis work is necessary for deciphering general patterns and trends

related to, e.g., site types, climate, and management, and the development of corresponding emission factors, i.e., estimates of the net annual soil GHG emission/removal, which can be used in GHG inventories. Development of specific emission factors also sets prerequisites for the background or environmental data to be reported in individual studies. We argue that wide applicability greatly increases the value of individual studies. An overall objective of this paper is to support future monitoring campaigns in obtaining high-value data. We analysed peer-reviewed publications presenting

$CO_2$, $CH_4$ and $N_2O$ flux data for drained organic forest soils in boreal and temperate climate zones, focusing on data that have been used, or have the potential to be used, for estimating net annual soil GHG emission/removals. We evaluated the methods used in data collection, and identified major gaps in background/environmental data. Based on these, we formulated recommendations for future research.

## 1 Introduction

Organic soils contribute to the atmospheric greenhouse gas (GHG) concentrations, as they can both remove and emit GHGs, and have globally extensive carbon (C) and nitrogen (N) stores (Post et al., 1982; FAO, 2012; IPCC, 2014; Oertel et al., 2016; Wilson et al., 2016). Organic soils are, especially in the boreal region, commonly peat, derived from plant remains that have accumulated below the high water-table (WT) of peat-forming wetlands, peatlands. Below the WT decomposition is anaerobic and generally slow (e.g., Straková et al. 2012). Peatlands have been widely used for peat

extraction or converted into agricultural and forestry land (Joosten 2010). These land uses typically involve drainage by ditching. Draining of organic soils enhances aerobic decomposition and thus the mobilization of their C and N stores (e.g., Post et al., 1985; Kasimir-Klemedtsson et al., 1997; Ernfors et al., 2008; Petrescu et al., 2015; Abdalla et al., 2016; Pärn et al., 2018). Forestry is an extensive land-use type on peatlands in northern Europe, especially in the Nordic and Baltic countries (e.g., Barthelmes et al. 2015). The drained organic forest soils of this region may act as significant sources

of GHGs (Barthelmes et al. 2015), and their annual carbon dioxide ($CO_2$), methane ($CH_4$), and nitrous oxide ($N_2O$) emissions and removals have to be reported in the national GHG inventories.

Currently, both the IPCC (2006) agriculture, forestry and other land use (AFOLU) guidelines and the IPCC (2014) Wetlands Supplement may be used for reporting the annual GHG emissions and removals for soils under anthropogenic land uses, such as drained organic forest soils. Area-based emission factors (EFs), describing the net annual soil GHG

emissions/removals, have been developed to reflect the impacts of ecosystem type, land management and environmental conditions. Countries may opt for different methodological levels in their GHG reporting, so-called Tiers 1 to 3, where Tier 1 is the simplest approach with default EFs of the IPCC. In IPCC (2014), separate EFs are available for on-site processes in soils, emissions from dissolved organic carbon (DOC) and other C-forms exported in leaching, and emissions from soil combustion (fires) for drained organic soils (Fig. 1). In practice, most countries currently use Tier 1 EFs for soil

emissions/removals by drained organic forest soils. These Tier 1 EFs are mean values of annualised net emission/removal estimates compiled from published data, categorized by climatic zones, and for some zones also by wide soil nutrient



status classes "poor" and "rich" (IPCC, 2014). Tier 2 and Tier 3 are methods that use country-specific data (Tier 2) and repeated forest inventories and/or advanced modelling (Tier 3), which should make the national estimates more accurate. Uncertainty of the estimated emissions is still high. For instance, the 95 % confidence interval for the Tier 1 $CO_2$-C EF

for boreal nutrient-poor soils ranges from -0.23 tonnes $CO_2$-C ha$^{-1}$yr$^{-1}$ removal to 0.73 tonnes $CO_2$-C ha$^{-1}$yr$^{-1}$ emission, and that for the corresponding $CH_4$ EF from 2.9 to 11 tonnes $CH_4$ ha$^{-1}$yr$^{-1}$ emission (IPCC, 2014). Even in Finland where a national Tier 2 method is used, the relative uncertainty of $CO_2$ emissions from organic soils in the reporting category 'forest remaining forest' is as high as 150 % (Statistics Finland 2019). This means that those soils can be either sinks or sources of $CO_2$, though the latter is more likely due to the estimated 1.1 Mt C decrease annually in the soil C stock of

those lands. The high uncertainty underlines the need for improvement of GHG emission/removal estimation in countries with a high proportion of drained organic forest soils.

Both data collection and method development for reporting the anthropogenic emissions from drained organic soils have duly received increasing attention with the aim to improve the accuracy of the emission estimates (e.g., IPCC, 2014; Oertel et al., 2016; Tubiello et al., 2016; Kasimir et al., 2018). The accuracy of EFs can be improved as more peer-

reviewed data become available and quantify a wider set of specific management options and ecological conditions for a given country or a region (e.g., Couwenberg, 2011). However, GHG emission data may be collected with several methodologies, which further involve different options for the detailed design of the measurement and data analysis protocols. Development of more specific EFs also sets prerequisites for the background or environmental data to be reported along with the GHG emission estimates in individual studies. Since collecting representative GHG emission data

is time consuming and thus costly, it would be beneficial to set certain guidelines for how to report the data and related environmental information, so that each individual study would also contribute to more general analyses based on data collation and modelling, or at least simple regression analyses on the factors potentially influencing the emissions. Yet, so far there has been no systematic assessment on how such data are presented in individual studies or how to improve the applicability of the data collected in individual studies in synthesis work aiming to develop more specific EFs. While

there is a growing number of long-term GHG measuring stations with standardized protocols (e.g., https://www.icos-ri.eu/), and a vision towards a more integrated approach (Kulmala, 2018), drained organic soils exemplify systems where many studies are still carried out with relatively small resources and manually operated systems. An overall objective of this paper is to assist the measurers-to-be of such campaigns to plan their data collection and presentation protocols, for enhancing the applicability of their data and thus getting the most out of their hard work.

We analysed peer-reviewed publications presenting GHG data for drained organic forest soils in boreal and temperate climate zones. We focused on data that have been used, or have the potential to be used, for estimating the net annual soil GHG emissions/removals for the measured sites. Such data can then be further used for constructing EFs. The emphasis was on emissions/removals of $CO_2$, $CH_4$ and $N_2O$ derived from biological processes taking place between vegetation, soil and the atmosphere on-site. We will henceforward call such annual emissions/removal estimates the "soil GHG

balance(s)". We set as our aims to i) collate a database that may be used for developing EFs, and ii) to examine:

- the data collection methods and data structure in the peer-reviewed publications potentially qualified for estimation of annual soil GHG balances,

- how the characteristics and applicability of the data produced by different GHG monitoring methods differ,

- which type of background data (e.g., tree stand and site characteristics) is provided in the publications that could be

used for generalizations and soil GHG balance modelling, and



- which information would be needed for the publications to provide improved applicability for generalizations and modelling.

Because of higher complexity in processes and monitoring approaches for the $CO_2$ flux, we will review $CO_2$ data in more detail than the other gases. Fire-induced emissions will not be dealt with. Water-borne C losses will be assessed to a
limited extent only, due to the scarcity of available data for drained organic forest soils, and a recent review published on the subject (Evans et al., 2016).

## 2 Material of the Review

We searched original studies in peer-reviewed literature on soil GHG exchange or C-stock changes in drained organic forest soils. IPCC (2014) Wetlands Supplement reference list contains most of the GHG flux data published until 2013,
and was used as a basis that was complemented using reference data bases, Web of Science, and Google Scholar.
From the retrieved peer-reviewed publications, we included in the database data that fulfilled the following specifications for site characteristics:

- Data were for organic soils. As criteria for 'organic soil' we followed Annex 3A.5 in IPCC (2006). This means thickness of the organic horizon greater than or equal to 10 cm, and a minimum of 12 % organic C by mass. In
practice this includes both C-rich histosols (peat) and other organic soils typically identified as gleysols, which have characteristically lower C-content than peat. If the publication did not specify the soil type or characteristics in an unambiguous manner, the data were excluded.
- Data were for forest soils. We followed specifications applied in IPCC (2014), where minimum criteria for forest canopy coverage are 10 per cent of the area and continuous forest area size more than 0.5 ha (as in FAO's FRA,
2015). To qualify as forest, the time passed since afforestation had to be over 20 years for sites previously under some other land use. The sites should have been under conventional management conditions, and thus sites with extreme experimental fertilization or hydrology manipulation were excluded. If the publication mentioned tree presence but did not provide sufficient information to confirm that the above forest criteria were met, the data were excluded.
- Data were for drained organic soils. Data were excluded, if it was not specified that the studied site was drained or drainage-impacted.
- Data were from the boreal or temperate climate zone as defined in IPCC (2006), and the monitoring/sampling location was detectable by coordinates.

We formed a database (S1 Table S1 and Table S2) for estimation of soil $CO_2$, $CH_4$ and $N_2O$ balances for drained organic
forest soils. The database includes publications released prior to year 2018 with data on (i) inventories integrating changes in soil C stocks, and (ii) $CO_2$, $CH_4$ and/or $N_2O$ fluxes monitored by (a) chamber technique or (b) eddy covariance (EC) technique. The few existing peer-reviewed soil $CO_2$ balance estimates based on EC data were assumed to be technically correct. In data derived by chamber methods, we paid attention to i) specification of the flux components monitored (i.e., total vs. autotrophic, ground vegetation presence or removal, inclusion of fluxes from the litter inputs above and
belowground), ii) temporal coverage to facilitate forming an annual estimate, iii) spatial coverage at the monitoring site, and iv) description of the methods in flux analysis. For soil inventory methods, we evaluated the ability of the chosen field-work method to provide representative samples with unambiguous references for determining the C-stock change over time. Further, to support constructing EFs and modelling of GHG emissions, available qualitative and quantitative



information on site characteristics was evaluated. Such information includes, e.g., temperature sum, site type (at least rich
/ poor), soil properties, WT regime, description of the forest stocking, tree species composition, and for afforested sites
the time of afforestation and previous land use.

### 3 Framework of the Review: the Processes and Structural Features to be Covered by Applicable Data

Quantifying the soil GHG balance, especially for $CO_2$, in forests growing on organic soils is technically challenging
because i) C-sequestration into plant biomass takes place in a voluminous and usually diverse vegetation community with
uneven spatial distribution, ii) the C transfer from biomass into dead organic matter as diverse litter forms takes place
both aboveground and belowground with uneven spatial distribution, the belowground transfers being especially
challenging to quantify, iii) physical and biochemical characteristics in organic soils change over time, iv) $CO_2$ release
through heterotrophic processes takes place both in recently deposited litter and in a soil composed of previously
accumulated dead organic matter, (v) $CO_2$ formed in the heterotrophic processes in soil must be separated from similarly
large $CO_2$ emissions formed in autotrophic root respiration in gaseous flux measurements, and vi) rates of biological
processes change over the year and differ between years depending on weather conditions, stand development and
management (points i-v shown in Fig. 1). In this paper "soil $CO_2$ balance" includes C transfer fluxes to the soil as above-
ground and belowground litter, and losses by decomposition of litter and soil organic matter.

The methods used to quantify soil $CO_2$ balance can be classified into gaseous flux monitoring methods and soil inventory
methods. The two method groups differ profoundly in the way they quantify the components of the soil C balance.
Multiple monitoring setups are available in both methods, which may influence the estimate formed. This should be
considered carefully when planning the measurements, because monitoring setups in most studies are chosen to provide
data for answering specific research questions, and they do not always aim to quantify the annual soil $CO_2$ balance. A
more detailed description of the methods, with their advantages, weaknesses and caveats, is given in supplement (S2).
The flux methods include i) EC flux monitoring by sensors located above the canopy, and ii) chamber techniques
involving chambers enclosing a known gas space over soil with or without ground vegetation, litter and roots. For
estimation of the soil $CO_2$ balance, data processing in flux-based methods usually requires additional data on mass-based
C stock changes, such as organic matter inputs as litter, or change in vegetation C stock. The soil inventory methods
integrate the outcome from all processes affecting the soil C stock over time. C in mass based C-stock change is converted
to $CO_2$ by multiplying with 3.67 (the mass ratio between $CO_2$ and C, 44/12).

For forming the EFs for $CH_4$ and $N_2O$ there is no guidance on how living vegetation presence or litter dynamics should
be taken into account in flux measurements, except that vegetation presence can be reported for $CH_4$ monitoring locations
(IPCC, 2014). However, wetland plants that have roots with aerenchymatous tissue, such as cottongrass (*Eriophorum
vaginatum*), a widespread sedge that is found also on drained sites, are known to pipe out $CH_4$ from waterlogged peat
layers (Askaer et al., 2011). Thus, excluding these plant types may lead to severe underestimation of the $CH_4$ flux (Askaer
et al., 2011); however, in drained sites sedges may also attenuate the emissions (Strack et al., 2006). Further,
methanotrophic symbionts dwelling in hyaline cells of *Sphagnum* mosses are able to oxidise $CH_4$ in solutes to $CO_2$ that
is consumed in photosynthesis (Raghoebarsing et al., 2005; Larmola et al., 2010). So far, such observations are available
for undrained peatlands only. There are also reports indicating that stem bark and leaves are able to transport $N_2O$ and
$CH_4$ from soil to the atmosphere in trees such as black alder (*Alnus glutinosa* L.) (e.g. Rusch and Rennenberg, 1998;
Gauci et al., 2010; Machacova et al., 2013; Covey and Megonigal, 2019; Welch et al., 2019), but the magnitude of such
tree-mediated pathways is still largely unknown. Furthermore, belowground biomass disturbance, e.g. rhizosphere and



mycorrhizal mycelia removal by trenching, has been shown to result in increased $N_2O$ flux in drained organic forest soils (Ernfors et al., 2011). We therefore paid attention to vegetation disturbance / removal when reviewing the $CH_4$ and $N_2O$

studies. It seems clear, however, that in future studies of $CH_4$ and $N_2O$, vegetation should be kept intact.

When estimating $CH_4$ fluxes, it is important to consider the drainage ditches (Fig. 1), which represent wet areas in a drained landscape, and may be local hotspots for emissions from ditch floor and the water column. GHG emissions can be released by diffusion through the water body, by ebullition and by gas transport through vegetation, e.g. sedges (Frenzel and Rudolph, 1998; Saarnio and Silvola, 1999; Natchimuthu et al., 2017), which need to be considered in monitoring.

Specific Tier 1 EFs have been constructed for ditch $CH_4$ emissions in IPCC (2014), where information on the proportion of the drainage ditch network area in the landscape is further needed for estimating the emissions. For further modeling of ditch emissions, information on drainage ditch water flow rates and levels, ditch characteristics, vegetation composition, and ditch network maintenance likely have importance.

**4 Availability of published data for soil GHG balance estimation**

We reviewed about 130 papers, and finally retrieved 52 studies that reported GHG fluxes or C-stock changes in drained organic forest soils in boreal and temperate zones with potential data for estimating soil GHG balance (S1). Several studies included more than one GHG species monitored (and thus, the total n of publications in Table 1 appears to be higher). Most of the $CO_2$/C studies used flux monitoring methods; however, studies using inventory methods covered, on average, more sites (Table 1). Studies on $CO_2$ had the highest total number of sites (133), while $N_2O$ monitoring studies had the

lowest (61).

The number of publications in our database (Table 1), complemented with more recent data, became notably higher than that in the IPCC (2014). Our database is not fully matching with the data included in the IPCC (2014) even concerning older data. This is firstly because some studies (8) in the former were replaced by newer publications using the same field data. Secondly, some publications did not match with our criteria, as described in the section 'Material of the Review'

(S1, Table S4). We identified each monitored site based on coordinates, site type, and other information provided in the publications, which prevented double-counting of sites that were, e.g., included in review papers. The number of $N_2O$ monitoring sites was further reduced by recent error detection for 40 sites (Ojanen et al., 2018).

Common reasons for exclusion of a study were insufficient descriptions of the monitored site and methods, unclear data presentation, or the same data found in multiple publications (S1 Table S4). Information about the soil type, forest

characteristics, or drainage status are important, and insufficient characterization may prevent a conclusion that the studied site represents drained organic forest soils. Unusual forest management conditions, such as experimentally applied unconventionally high amounts of lime or fertilizers, restrict data inclusion from such monitoring sites.

Somewhat more difficult question is how to deal with data quality. Data quality remains undefined if the design of spatial and temporal extent of soil sampling or flux monitoring, or the analytical procedures in the laboratory are not clearly

described. Whereas the EC method is expected to integrate the C balance over a large area around the sampling spot, absence of spatial replicates on the heterogeneous forest floor in the chamber and soil inventory methods raises concern regarding the representativeness of the monitoring setup. Another concern in flux monitoring by chambers can be a low sampling frequency and/or extent over time. Conditions in the environment, e.g. vegetation, soil temperature and WT change over time, and need to be included in monitoring not only during the warm season but also during shifts from/to

colder seasons. It becomes also overly challenging to estimate cumulative seasonal or annual fluxes if data are presented as series of daily flux values, daily mean flux values or a range of flux values. Some of the methods are no longer



considered to produce reliable results, e.g., soda lime absorption for $CO_2$ flux estimation in field conditions (see S1 Table S4, S2).

**5 Applicability of the published data for soil GHG balance estimation**

**5.1 Carbon dioxide**

**5.1.1 Chamber methods**

Flux data monitored by dark chambers forms the largest data set for forests on drained organic soils (Fig. 2, S1). However, complete soil $CO_2$ balance estimates based primarily on data collected on-site are rare (Ojanen et al., 2010, 2013; Meyer et al., 2013; Uri et al., 2017). Ideally, a setup for forming the soil $CO_2$ balance by dark chamber techniques would include
quantification of the heterotrophic emission sources (litter and soil) without autotrophic emissions from live plants, but in reality this condition can be achieved only by models on data derived from multiple monitoring setups as explained in S2 (see also Ojanen et al., 2010, 2013). Pavelka et al. (2018) provides minimum requirements and recommendations for GHG monitoring of terrestrial ecosystems by chambers.

Generally, cooler night-time temperatures result in lower emissions (Brændholt et al., 2017). Not accounting for this
pattern results in overestimated emissions. Automated gas flux monitoring with short intervals ensures capturing the impact of diurnal soil temperature differences on $CO_2$ emissions. Diurnal $CO_2$ flux monitoring by automated chambers has been deployed in two studies (Ball et al., 2007; Meyer et al., 2013). In manual chamber data, the diurnal temperature differences have been taken into account mostly by applying temperature modelling into fluxes monitored during day-time in the boreal zone studies. However, only 36 % of the temperate zone studies accounted for diurnal temperature
differences by collecting flux data also during night-periods or by modelling (S1). Consideration given to soil temperature impacts on GHG fluxes should be a requirement in data collection, processing and reporting in studies using manual GHG flux data collection.

Soil C balance is the balance between C added in litter inputs and C lost as $CO_2$ in emissions from litter and soil organic matter decomposition. The most typical data lacking for completion of the soil $CO_2$ balance estimate in the reviewed
publications was the annual rate of litterfall (Fig. 1 & 2, S1 Table S1). Emissions from decomposing litter are included in $CO_2$ flux monitoring by having the deposited litter on the soil surface intact, but even then the rate of litter inputs need to be measured, or estimated, to complement the balance. In studies where the monitored surfaces are kept clean from litter, the above-ground litter $CO_2$ emission must be estimated separately, which may be laborious and result in bias or error. Extensive studies on annual aboveground litter production and decomposition with impact assessment to soil $CO_2$ balance
have been made for the boreal zone in Finland (Ojanen et al., 2013, 2014). Comparable integrated assessments for the temperate region a for afforested sites, formerly used for peat mining or as cropland, are still lacking. Tree species-specific aboveground litter production estimates are available for birch, pine and spruce, if measures quantifying the tree biomass are known (e.g., Repola 2008, 2009). Considerably less specific data are available on understory litter production (Straková et al., 2010), litter decomposition (Domisch et al., 2000; Tuomi et al., 2010; Straková et al., 2012; Ťupek et al.,
2015), and, especially, on belowground (fine root) litter production and decomposition rates (Laiho et al., 2003, 2014; Finér et al., 2011; Jagodzinski et al., 2016; Bhuiyan et al., 2017). Use of generic values for litterfall and litter decomposition cannot be recommended because these rates are site-type specific, typically differing between nutrient-poor and rich sites, and also depend on growing season length (Straková et al., 2010, 2011, 2012; Ojanen et al., 2013;



Lehtonen et al., 2016). For more accurate soil $CO_2$ balance estimates, work towards reduced uncertainty in the inputs and
decomposition rates of different litter types under different conditions is needed.

Above- and belowground autotrophic respiration of vegetation remaining inside the chamber is a $CO_2$ flux source that was often acknowledged but not always quantified in the dark chamber studies (Fig. 2, S1 Table 1). For estimation of soil $CO_2$ balance, $CO_2$ emissions both from canopy and roots of the ground vegetation remaining inside the monitoring plot are practically impossible to quantify afterwards, and thus should be given consideration when performing the
measurements and reporting the results. In some studies the soil surface has been free of ground vegetation either naturally due to shading by tree canopy, or kept free by frequent clipping. The living roots of both ground vegetation and trees extending to the flux monitoring plot may still form an autotrophic $CO_2$ emission source unless specifically excluded by trenching (Subke et al., 2006). Although an approximately 50 %-proportion between total and autotrophic respiration is a fairly common outcome in studies conducted on both organic and mineral soils (e.g., Bond-Lamberty et al., 2004;
Comsted et al., 2011), use of a literature-based fixed coefficient induces a source of uncertainty with a potentially high impact on the soil $CO_2$ balance estimate. For example, in Uri et al. (2017) the proportion between heterotrophic and total soil respiration was between 0.6 and 0.7 in five downy birch (*Betula pubescens*) stands on nutrient-rich drained temperate peatlands. The soil $CO_2$-C balances in Uri et al. (2017) would differ on average by 54 % from the reported estimate if the simple 0.5 proportion between the heterotrophic and total soil respiration fluxes were used in the calculus. Modelling,
based on on-site flux monitoring, has been done to separate the autotrophic and heterotrophic soil respiration for drained boreal peatlands with varying nutrient status, tree species composition and age (Ojanen et al., 2010, 2013) and for nutrient-rich temperate peatlands with downy birch (Uri et al., 2017). Both these studies indicated that annual total and heterotrophic respiration both correlate with soil temperatures, but there is substantial between-site variation in the annual fluxes. The share between heterotrophic and autotrophic respiration further varies over the growing season depending on
the phenology of trees and understorey vegetation, and this introduces another source of uncertainty in data where heterotrophic emission is proportioned from the monitored total soil $CO_2$ flux. Data quantifying both total and autotrophic respiration for sites potentially influenced by preceding management impacts, such as afforested former peat mining areas or croplands on organic soils, are currently not available at all. Use of original site-specific heterotrophic emission data integrates local environmental conditions best and should be quantified in flux monitoring. However, it would be useful
if general models estimating the proportion between total and autotrophic soil respiration in different types of forests in open, maturing and mature stages, in conditions created by recent management, and with seasonal impacts were further developed.

Several studies monitored $CO_2$ exchange using transparent chambers, but their flux estimates were only rarely suitable for estimating soil $CO_2$ balance. The advantage of transparent chambers is that they can be used for estimating ground
vegetation C balance. However, a complication of the method in forests is separating tree root respiration from ground vegetation fluxes. Trenching may be done to cut out roots reaching inside the monitoring plot from outside, but this may also disturb the ground vegetation inside the plot, especially clonal plants for which the rhizomes may extend far beyond the plot limits. That is why this method may produce quite ambiguous results if applied in forests.

### 5.1.2 Eddy covariance method

Eddy covariance (EC) method was applied in three studies (Lohila et al., 2007, 2011; Meyer et al., 2013). The EC method yields the net $CO_2$ exchange (NEE) between the ecosystem and the atmosphere, and a set of calculations as well as additional measurements are needed for producing the soil $CO_2$ balance (S2, Lohila et al., 2007, 2011; Meyer et al., 2013).



EC data combine typically high temporal flux sampling intensity with a large areal coverage, i.e. the data has good representativeness for the studied area, and has a relatively small standard error in the NEE estimate (Lohila et al., 2011).

For estimating soil C balance as 'NEE minus change in vegetation biomass'(S2), the greatest biomass change in forested sites is naturally in the tree stand. If ground vegetation biomass is low under closed canopy conditions, it can be neglected. The tree biomass change data are usually based on systematic stem radial growth and height growth measurements providing cumulative annual data that is combined with biomass allocation models. The errors can be propagated as in Lohila et al. (2011), in which study they were 11.4 % of the mean for NEE and 20 % for annual tree biomass increment.

For estimating soil $CO_2$ balance, combining flux data from EC and automated chambers did not result in lower error compared to the method where mass based biomass change data were applied (Meyer et al., 2013).

### 5.1.3 Inventory methods

Inventory methods were applied in five studies (Minkkinen and Laine, 1998; Minkkinen et al., 1999; Simola et al., 2012; Pitkänen et al., 2013; Lupikis and Lazdins, 2017). On average, studies using inventory methods included a higher number

of sites monitored compared to studies using flux methods (Table 1). In the largest study, the soil C-stock change estimates of 273 peatland sites were pooled into groups representing three site types for five regions in Finland (Minkkinen and Laine, 1998), while site specific estimates were given in the other studies. In this method, soil C stocks are measured or estimated at least twice. The C-stock difference is calculated from dry bulk densities and C concentrations in volumetric soil samples taken from the peat surface down to the bottom of the peat deposit. Alternatively, sampling may be extended

down to a clearly definable reference layer. Soil $CO_2$-C balance estimates based on inventory data integrate the outcome from all C stock contributing processes over long (decadal) periods. Thus, the method is good for monitoring soil C stock differences over time in stabilized conditions, but the drawback is the difficulty in determining a small temporal change in a very large soil C stock (e.g. Minkkinen and Laine, 1998). Year-to-year differences in soil C stock or specific forms of C or GHGs cannot be studied, which limit the use of the method only for Tier 1 EFs. Reliable estimates may be obtained

only if the bottom of the peat deposit is defined accurately and in a similar manner in the repeated sampling, or if an unambiguous reference layer is used that is located deeper in the soil profile than the depth to which anthropogenic changes may be expected to extend (S2). These conditions cannot usually be met, and thus the inventory methods usually involve very high variation around the estimates, which is further contributed to by the spatial variation in peat characteristics and the topography of the bottom of the basin, since exactly the same spots cannot be resampled.

**5.2 Methane and nitrous oxide**

$N_2O$ and $CH_4$ fluxes have been studied specifically or together with $CO_2$ flux monitoring (S1 Table S1 and Table S2). Most studies (90 %) on $CH_4$ and/or $N_2O$ fluxes used measurement points where ground vegetation, if present, remained intact inside the chamber. In three studies vegetation from the flux monitoring surfaces was regularly removed (Danevcic et al., 2010; Ernfors et al., 2011; Holz et al., 2016). In combined $CO_2$, $CH_4$ and $N_2O$ monitoring plots, where surface

vegetation and litter is removed and/or soil is trenched for studying the heterotrophic $CO_2$ flux, the caused disturbances in vegetation and soil conditions may influence the $CH_4$ and $N_2O$ fluxes. In Tier 1 EFs by IPCC (2014) only climate, land use and soil nutrient status information is used. Presence or absence of plant species that are able to transport $CH_4$ can be accounted in a Tier 2 method, but there is no specific guidance for how to stratify. For Tier 2 EF's guidance and clarification on how to include ground vegetation, rhizosphere and litter would be useful. For constructing Tier 2 factors

it should be recommended in any case that ground vegetation should be kept intact in $CH_4$ and $N_2O$ monitoring.





Drained organic forest soils are often small annual sinks of $CH_4$ (Minkkinen et al., 2007a; Ojanen et al., 2010). Yet, ditches in such sites may yield significant emissions. A relatively small number of studies have quantified the contribution of $CH_4$ fluxes in forests (Roulet and Moore, 1995; von Arnold et al., 2005b; Minkkinen and Laine, 2006; Glagolev et al., 2008; Sirin et al., 2012). To our knowledge only Peacock et al. (2017) has measured $N_2O$ emissions
ditches at semi-natural and cropland sites, and found them to be significant. Additional flux data are therefore needed for quantification of this flux in drained forests. To increase applicability, publications on ditch GHG emissions should also provide information on the ditch characteristics, such as size, spacing, current maintenance regime, water level and flow rates during monitoring, and vegetation.

**5.3 Water-borne C**

Both pristine and drained organic soils show C losses in drainage waters (e.g., Strack et al., 2008; Urbanová et al., 2011; Nieminen et al., 2015). Water draining from organic soils contains dissolved organic C (DOC, typically defined as C passing through a 0.45 μm membrane filter) and particulate organic C (POC), the sum of DOC and POC being total organic carbon (TOC). To estimate water-borne C fluxes, quantification of water flux as well as C concentrations is necessary. Also, incoming C in precipitation and influx from surrounding forest soils should be accounted for, when
estimating net water-borne C fluxes. There are several publications reporting the C concentrations in waters, but complete water flux estimates for a specific forest area/catchment are rare (S1 Table S3). In practice, the water fluxes are often estimated using models. Current data are too limited for forming explicit views about data applicability. For advancing the knowledge base, it would be useful to include site specific characteristics and climatic conditions in data collection and reporting, such as size and location of forest in the catchment topography, ditch depth and spacing, vegetation type,
and annual precipitation characteristics. Waterborne carbon loss on drained peatlands is included in the review by Evans et al. (2016).

**5.4 Reporting of key drivers for soil GHG balance**

We currently have the understanding that the GHG fluxes generally depend on site nutrient status, size and characteristics of the tree stand, soil temperature, and the WT regime (von Arnold et al., 2005a, 2005b; Ojanen et al., 2010, 2013, 2014).
These parameters are not, however, routinely reported in studies quantifying GHG fluxes (Table 2).

In the reviewed data, focusing on forests on drained organic soils, a surprisingly large proportion of the papers failed to provide information on the basic characteristics of the tree stand (Table 2). Stand volume was the most commonly reported parameter, but still only in 50 % of the studies. The tree stand may influence the soil GHG balance in several ways. A large stand volume lowers the WT through canopy interception of precipitation and evapotranspiration (Sarkkola et al.,
2010), which may lead to high $CO_2$ emissions and a soil sink of $CH_4$ (Minkkinen et al., 2007a; Ojanen et al., 2010). Different tree species produce litters of different quality (e.g., Straková et al., 2010), which decompose at different rates (e.g., Straková et al. 2012) and have been found to result in differing soil GHG fluxes on mineral soils (e.g., Papen and Butterbach-Bahl, 1999; Butterbach-Bahl et al., 2002). Further, tree stand information may be needed for estimating tree litter inputs if those have not been measured.
The volume of increasingly oxic soil above the WT is important for aerobic decomposition processes producing $CO_2$. Also for the balance in processes producing and consuming $CH_4$ in soil, i.e. methanogenesis and methanotrophy, the WT depth influence on oxic and anoxic soil environment is critical. Data provided on WT depth and dynamics were often either lacking, were presented as line graphics describing annual or seasonal WT, or were provided for the day of flux





monitoring or as average over a longer period. Average annual or seasonal WT were provided in less than half (44 %) of
the publications (Table 2). This lack of applicable WT data seriously hampers using this data for meta-analyses and
development of more dynamic EFs. Having both mean annual WT and more detailed WT characteristics (e.g., monthly
mean and median, quartiles for growing season, frost free period and year) in the publication would allow inspection of
soil GHG fluxes in specific conditions, such as comparisons between shallow-drained and deep-drained (WT≤30 cm vs.
WT>30 cm from the soil surface; IPCC 2014) conditions.

Less than a third of the publications reported physical (e.g., bulk density) or chemical characteristics (e.g., C, N, and P
concentrations, pH) of the soil (Table 2). Moreover, differences in the extent of surface soil layers sampled in the studies
reduce data comparability. Chemical quality of the organic matter is known to constrain its decomposition rate (e.g.
Straková et al., 2012) and the resulting GHG fluxes. Site type, CN-ratio and bulk density have been found to correlate
with heterotrophic $CO_2$ emission (Ojanen et al., 2010), whereas $N_2O$ flux increases with lower peat CN-ratio
(Klemedtsson et al., 2005; Ojanen et al., 2010, 2018; Pärn et al., 2018). To some extent, this soil quality aspect is taken
into account in IPCC (2014) Tier 1 level EFs by using 'nutrient-poor' and 'nutrient-rich' site categories, based largely on
the origin of nutrients, i.e. poor sites receiving nutrients from the atmosphere only and rich sites receiving nutrients also
from other sources (ground water and flood water). For generalization of GHG fluxes in different site conditions and
organic soil types, e.g. by model development, concentrations of the key elements (C, N, P) that are part of the
decomposition process should preferably be included in reporting. Sampling depth for determining soil characteristics in
drained forest soils should be within the vegetation rooting zone and above the WT. A 0–20 cm soil layer was the most
commonly used, and would be an easy standard as it does not require very specific sampling tools like deeper coring.
Both the rooting zone and the WT are often deeper than this, however, and a specific study might lead to a better-motivated
standard. Also, it should be noted that on long-drained sites, there may be a thick surface layer accumulated from post-
drainage tree litter (e.g., Saarinen and Hotanen, 2000; Straková et al. 2010), corresponding to mineral soil forest O and H
horizons. Such layer may be difficult to separate from the actual peat soil (Laiho and Pearson, 2016), but should also
better not be separated, since it is also affecting the soil GHG fluxes and balance.

All studies commendably reported the coordinates of the sites. This is important since coordinates unambiguously specify
site location, and, e.g., allow retrieval of climate data.

**5.5 Spatial scale covered with different methods**

Spatial scale varies for methods used for GHG and soil C-stock change monitoring from point measurements (peat cores)
in inventory methods, to ca. 0.5–1 $m^2$ in plots monitored by chambers, and further to a flux source area (footprint area)
of over thousand square meters in the case of EC monitoring. An increase in the number of spatial replicates, i.e. the
number of monitoring points, increases the spatial representativeness in both inventory and gaseous flux monitoring by
chambers. In the reviewed soil inventory studies, multiple-site surveys included 1–5 sampling points at each site and 1–
3 replicate cores at each sampling point. In studies utilizing chamber techniques, on an average there were 8 replicate flux
monitoring points per site for $CO_2$ (range 2 to 48), 5 for $CH_4$ (2 to 16) and 5 for $N_2O$ (2 to 16). The size of flux monitoring
points varied from 10 cm diameter areas monitored by cylindrical chambers to 60 cm × 60 cm areas enclosed by
permanently installed frames. It can be reasoned that one EC tower gives an integrated flux for the whole footprint area,
while the representativeness in flux estimates based on chambers can be limited if common site vegetation, soil, or
topography characteristics are not covered by the monitoring points, and/or if the areal proportions of these properties are
unknown. On the other hand, the closed chamber technique is the best option for studying GHG fluxes from (small-scale)





specific soil surfaces, and is often used to complement EC monitoring. In soil inventory methods as well, attention to representative sampling at the study site is important. This can, however, not always be realized, as the repeated sampling

needs to follow the initial sampling design of the former study in the past, which has typically been designed for other purposes (Minkkinen and Laine, 1998; Simola et al., 2012).

Sampling procedures are strongly constrained by resources and are often trade-offs between spatial and temporal representativeness. It has not been thoroughly investigated so far, how the spatial and temporal measurement frequencies affect the precision of the estimated soil GHG balance. Such an analysis would be beneficial for structuring measurements

towards better landscape-level soil GHG budgets, and such analysis could be based on, e.g., data from sites where both EC and chamber methods have been applied.

**5.6 Temporal scale covered with different methods**

The temporal scale of GHG flux sampling ranges from continuous sampling with EC, to automated chamber monitoring at varying frequencies, and non-continuous manually performed (day-time) sampling from chambers in intervals of

several days to weeks. If GHG flux data collection is continued over several years, the multiple annual soil GHG balance estimates obtained yield a valuable description of the dynamics of the GHG fluxes in varying environmental conditions. In about half (53 %) of the flux studies GHG monitoring lasted for at least 2 years, and thus nearly half of the publications included data from one-year or shorter monitoring. Most studies (77 %) included also at least some flux monitoring events during cold (winter/frosted soil/snow cover) periods, while a small (7 %) proportion of the studies were restricted to the

'warm season'. Such 'seasonal' flux data collection periods were described, for example, as 'snow-free period', 'warm season', or a period between two specific dates, which does not provide an unambiguous or easy way to extrapolate the results of the monitoring period to the rest of the year. The IPCC (2014) applied an annualization coefficient of 1.15 for the few 'seasonal' GHG flux estimates that excluded the cold period. This coefficient was formed for boreal and subarctic climate regions on the basis of studies in which both warm and cold season GHG fluxes have been quantified (Dise, 1992;

Aurela et al., 2002; Kim et al., 2007; Alm et al., 1999b; Leppälä et al., 2011). Use of such a fixed coefficient is a source of uncertainty, since i) the length of the (un)monitored period may vary from study to study, and ii) 'seasonal' flux data and data used for forming the coefficient may not come from comparable climatic or site conditions. Although winter-time fluxes form a relatively small proportion (15 % as applied in IPCC, 2014) of the annual flux, more year-round empirical field data from a larger number of sites in drained conditions would be beneficial for further modelling of cold

season GHG fluxes. This may be especially critical for regions where the frost-free part of the cold season is lengthening, which may well affect the soil GHG balance.

At least $CO_2$ flux correlates even with small changes in the topsoil temperatures (e.g. Brændholt et al., 2017), and thus flux monitoring over time for cumulative flux estimates should include diurnal and annual temperature conditions reliably (Sander and Wassmann, 2014), and ideally should be as continuous as possible over seasons. If automated chamber

monitoring is not possible, irregular and diurnally imbalanced GHG flux monitoring data should be corrected by soil temperature – GHG flux relation.

Soil inventory methods, in contrast to flux monitoring methods, integrate all soil C stock changes (C losses as $CO_2$ and $CH_4$, water-borne C losses, and new C accumulation from litter inputs) over time periods of years to decades in the past into one soil $CO_2$-C balance estimate. Thus, inventory studies done with sufficient spatial coverage and accuracy in

determining the boundaries of the studied layer would give the most robust estimates on soil $CO_2$-C balance, especially when carried out over a period with no major land-use or environmental changes. When several land-use or management



changes have taken place during the time period covered by the repeated sampling, the average soil $CO_2$-C balance obtained may not describe any specified condition. Thus, it may be concluded that generally, estimates obtained by flux methods are better suited for GHG inventories aiming to report current fluxes and their dynamic responses to

management. Having said that, GHG flux studies on the impacts of typical management events (e.g. thinning, clear cutting, draining improvements) or covering a complete forest rotation cycle (open, maturing and matured stages) are yet to come.

## 6 Summarizing conclusions on data and further data needs

Basic definitions and guidelines for forming EFs for GHG inventories on organic soils are provided by IPCC (2006,
2014). Datasets used for forming EFs have passed peer reviewing during the publication process, and later evaluation by expert teams, but there are no guidelines for the data content and reporting. Consistent data would increase the applicability of the data for forming more specific Tier 2 EFs, and in other synthesizing assessments. We have identified issues in data content and reporting that have potential to further increase applicability of the data for these purposes. Each data collection method and data type has its strengths and weaknesses that contribute to the final outcome when
converted to soil GHG balance estimates. It would be highly beneficial to consider post-publication data use already during reporting by providing details on site characteristics and conditions, relatively easily acquirable measurements that have potential to correlate with GHG fluxes (Table 2). We identified major gaps in data, and provide some development suggestions for future data collection, as follows:

- Lack of applicable data, mostly due to a lack of environmental data, hampers developing more dynamic EFs than
480        mere averages that currently provide the most basic Tier 1 level for GHG inventories.
- More details on the characteristics and conditions at the monitoring sites (Table 2) are necessary to better analyse and synthesize the general dependencies between the GHG fluxes and environmental parameters.
- GHG and environmental data collection should cover whole forest rotations, by selecting comparable sites representing different stages of stand development for monitoring, as most of the current data are primarily snapshots
485        covering a few years at best.
- Consideration given to diurnal and longer term soil temperature impacts on monitored GHG fluxes should be a requirement for manual GHG flux data collection by chambers.
- More empirical cold season GHG flux data is needed for modelling.
- Flux monitoring period restricted to 'seasonal' (warm period) monitoring should start and end by defined weather
490        conditions and dates, e.g. presence of soil frost, growing season, which would help in extrapolating the results of the monitoring period to the rest of the year more consistently in modelling.
- There is a lack of studies relating GHG fluxes and long-term WT regimes (e.g., shallow drained vs. deep drained conditions) and of unambiguous water table summaries in GHG flux reporting in general.
- General models estimating the contribution of autotrophic respiration to the total soil $CO_2$ emissions in different
495        types of forests at different stages of stand development should be developed.
- Work toward reduced uncertainty in production and decomposition rates in belowground litter types, e.g., fine roots, in different conditions is needed because these data are still only sparsely available and typically not quantified in flux studies.



- There is a need for integrated studies on annual aboveground litter production and decomposition with impact assessment to soil $CO_2$ balance for the temperate region and for afforested sites, formerly used for peat mining or as cropland.
- The indirect short- or longer-term impact on GHG fluxes of forest management events, such as clear cutting, thinning and ditch network maintenance, should be quantified.
- $CH_4$ and $N_2O$ fluxes from trees should be quantified, for different tree species under different WT regimes.
- In future studies of $CH_4$ and $N_2O$ fluxes, vegetation and litter should be kept intact in the flux measurement points.
- $CH_4$ and $N_2O$ flux data quantifying emissions from drainage ditches are needed for different site types and ditch conditions. Ditch characteristics should be reported for the monitored sites.
- Current water-borne C-flux data are very limited, and thus there is need for data quantifying these C-fluxes from drained organic forest soils.

*Author contributions*. All authors planned the research jointly and contributed to data collection. JJ retrieved and reviewed the data, compiled the database, wrote the first draft of the manuscript, compiled supplementary information, coordinated the commenting and revisions that were provided by all authors, and compiled the following versions together with RL. KM drafted the description of inventory method (S2.1), ALo the description of EC method (S2.2.), and PO the description of chamber method (S2.3), which are presented in the Supplement.

*Acknowledgements*. The study is part of SNS-120 project 'Anthropogenic greenhouse gas emissions from organic forest soils: improved inventories and implications for sustainable management' funded by Nordic Forest Research (SNS). SNS-120 is a spin-off of the CAR-ES network also funded by SNS. This study was further supported by the Academy of Finland (grant 289116), Ministry of Education and Research of Estonia (grant PRG-352), Danish Innovation Fund for the Facce-Eragas project INVENT (grant 7108-00003b), University of Helsinki grant to 'Peatland Ecology Group', and the European Union through the Centre of Excellence EcolChange in Estonia.

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



**Table 1. Number of sites and publications estimating annual soil balances of $CO_2$, $CH_4$ and $N_2O$ for drained organic forest soils in boreal and temperate zones in this study and in the IPCC (2014) Wetlands Supplement.**


| GHG | Method | This study | | IPCC 2014[1] | |
|---|---|---|---|---|---|
| | | n–sites | n–publications[2] | n–sites | n–publications[2] |
| $CO_2$ | Inventory | 45 | 5 | - | - |
| | Flux (chambers) | 85 | 19 | - | - |
| | Flux (eddy covariance) | 3 | 3 | - | - |
| | Total | 133 | 27 | 133 | 13 |
| $CH_4$ | Flux (chambers) | 101 | 32 | 143 | 22 |
| $N_2O$ | Flux (chambers) | 61 | 31 | 131 | 20 |
| [1] Data from the IPCC (2014) Wetlands Supplement Tables 2.1, 2.3 and 2.5. [2] Some publications include estimates for multiple GHGs. | | | | | |





**Table 2. Potential GHG flux drivers, and respective information availability for the monitored sites in the reviewed published 52 soil GHG flux studies on drained organic forest soils.**

| | Measure | Papers [1] | Possible relation in soil GHG fluxes in larger data analysis |
|---|---|---|---|
| **Management** | Time of site draining | 38 (73 %) | Describes land management duration as (forestry) drained site. May affect GHG fluxes since length of time during which efficient aerobic decomposition of surface peat has taken place may affect peat characteristics. |
| | Management history described | 52 (100 %) | Draining improvements, fertilization, thinning, selective logging and other operations conducted in known time periods in the past may have influence on soil GHG balances. |
| | Ditch spacing and characteristics described | 1 (2 %) | Indicates draining conditions and is useful for assessing ditch GHG emissions. |
| | Ditch maintenance condition described | 0 (0 %) | – " – |
| **Tree stand** | Volume<br>Basal area<br>Stem number<br>Stand age | 26 (50 %)<br>5 (10 %)<br>7 (13 %)<br>16 [2] | Describes forest above-ground C stocking and litter input capacity, correlated with WT through rain interception and evapotranspiration. |
| | Species composition | 52 (100 %) | Deciduous / conifer dominance, or mixed forest structure may produce aboveground litter types with differing characteristics and thereby influence decomposition. Different species may also have differing transpiration rates, affecting WT. |
| | Productivity | 1 (2 %) | Classification based on expected tree growth potential to 'typical' and 'low productivity ' sites, where the latter includes sites with characteristically low forest stand stocking and growth due to nutrient deficiency, nutrient imbalance or hydrological conditions (despite draining), and this has impact on soil GHG balances (as in IPCC 2014). |
| **Site and soil** | Site type [4] | 50 (96 %) | Similar sites (by vegetation characteristics, soil nutrient status and characteristics) likely have similarities in GHG dynamics, and this is useful in grouping sites into similar categories. |
| | Ground vegetation composition and cover | 32 (62 %) | Indicator of soil fertility, moisture and shading conditions, and important for decomposition activity in soil. |
| | Presence and proportions of different plant functional types in the ground vegetation | [5] | Simple classification based on ground vegetation dominance by shrubs / herbs / grasses likely indicate soil nutrient status, thereby possibly influencing decomposition, and this classification can be practical for grouping sites into similar categories. |
| | Pre-drainage ombrotrophy or minerotrophy | 52 (100 %) | In general, peats of ombrotrophic and minerotrophic sites differ in soil quality and decomposition activity. |
| | Soil type | 52 (100 %) | Peat and other organic soil types (gleysols, muck etc.) differ by formation and characteristics, which may influence soil GHG balances. |
| | Organic soil thickness [3] | 29 (56 %) | Shallow organic soil may be impacted by minerogenic waters and mineral soil underneath, and thus have higher decomposition activity than deeper organic soils. |
| | Soil bulk density | 26 (60 %) | High bulk density values may indicate presence of mineral substrates, non-peat soils and/or possible disturbance in organic soil layer, which may influence soil GHG balances. Bulk density is also correlated with the degree of decomposition (e.g., Päivänen, 1969; Silc and Stanek, 1977) and water retention characteristics of the peat (e.g., Weiss et al., 1998), which may affect the GHG fluxes. |
| | pH<br>C<br>N | 30 (58 %)<br>18 (35 %)<br>24 (46 %) | Topsoil nutrient status and pH may influence vegetation composition, rate of C sequestration by tree stand, litter quality, |





| | | | |
|---|---|---|---|
| | C/N-ratio<br>P | 24 (46 %)<br>14 (27 %) | and decomposition rate. Peat layers for which data have been given also vary. A common standard could be 0–20 cm layer. |
| **Drainage** | Average WT levels in soil:<br>• Annual<br>• Warm season [6]<br>• Cold season [6] | <br>23 (44 %)<br>4 (8 %)<br>0 | WT level has major impact on decomposition processes and $CH_4$ production and oxidation rates, and thus basic WT characteristics would be useful to summarize in numeric form, e.g., monthly mean and median, and also quartiles for growing season, frost free period and year. |
| **Climate and weather** | Average air temperatures:<br>• Annual<br>• July<br>• February<br>• Monitoring period [6] | <br>34 (65 %)<br>9 (17 %)<br>9 (17 %)<br>0 | Air temperature has impact on litter production and topsoil decomposition processes. Inter-annual differences in air temperatures are potentially useful for modeling and detecting weather extremes during measurements. |
| | Annual air temperature sum | 7 (13 %) | Describes the temperature climate and annual conditions in a cumulative manner. |
| | Average soil temperatures:<br>• Annual<br>• Monitoring period [6] | <br>5 (10 %)<br>0 | Topsoil temperatures influence especially aerobic decomposition processes and are influenced by diurnal air temperature, and temperatures below the WT influence anaerobic decomposition processes. |
| | Precipitation:<br>• Annual<br>• Warm season [6]<br>• Cold season [6] | <br>32 (62 %)<br>0<br>0 | Cumulated precipitation may influence decomposition processes in soil (form a proxy for soil wetness or dryness). |

[1] Number (and proportion) of papers included in the database that provide the specified information

[2] For planted sites time of planting given with precision from year to a decade

[3] Specific or average values for the monitoring site soil characteristics, not minimum/maximum or a range

[4] Site type based on defined generally applied classification system

[5] Not countable from the papers in unambiguous way for comparisons

[6] Not countable from the papers in unambiguous way as the data collection periods were described, for example, as 'snow-free period', 'warm season', or as a period between two dates



**Figure 1: $CO_2$, $CH_4$ and $N_2O$ fluxes and mass transfer components contributing to soil C-stock changes in a forest ecosystem on drained organic soil, as in IPCC (2014). Arrows indicate flux/transfer direction.**

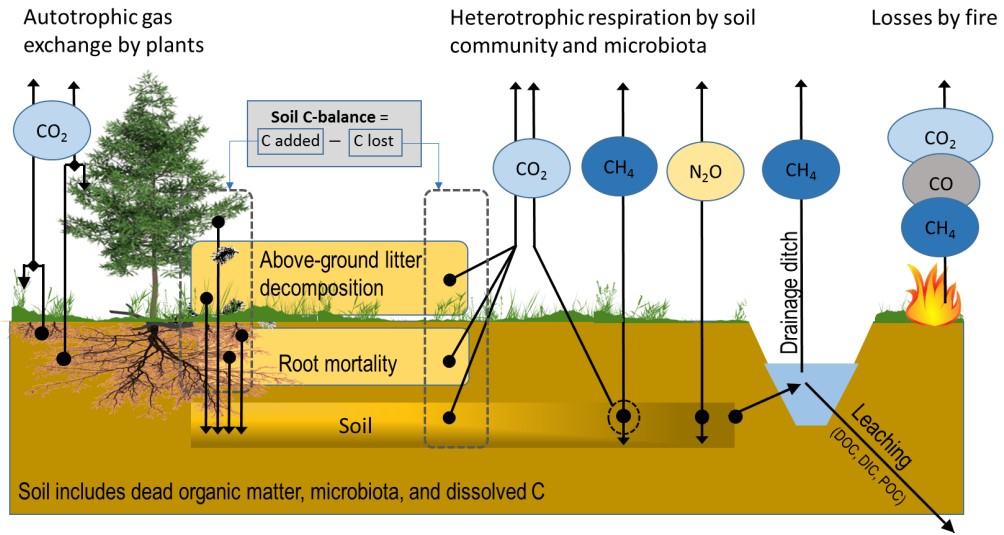





**Figure 2: Forest C stock, processes resulting in changes in the dead organic matter C stock in soil, C fluxes typically monitored**
**(see also S1 Table 1), and complementary data sources needed for forming soil CO₂ balance estimates (black arrows) in incomplete flux monitoring setups, according to IPCC (2014). Numbers I-IV next to monitoring setups by dark chambers refer to respective studies listed in S1 Table 1. Water-borne C losses and losses by fires are excluded from the figure.**

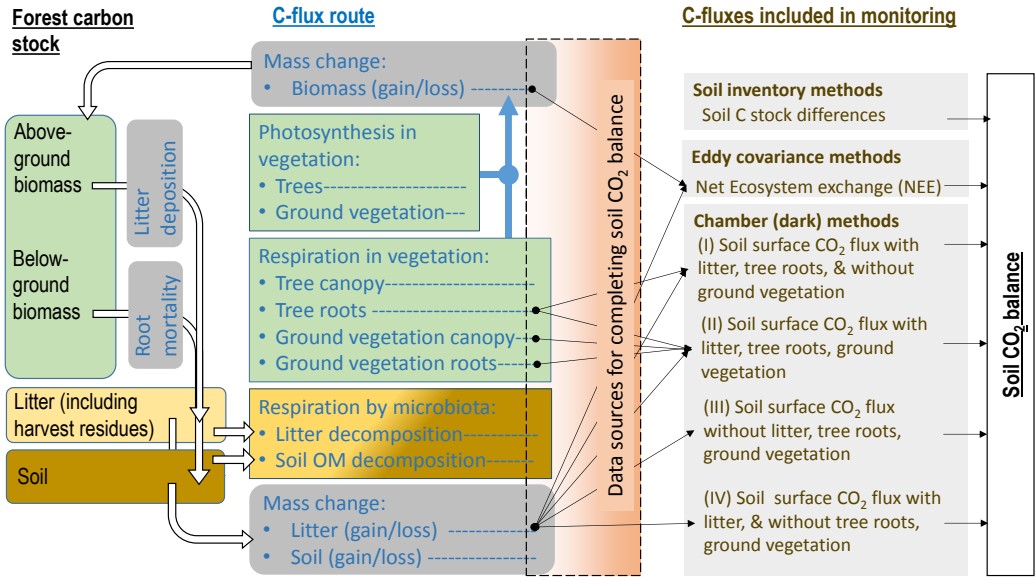