# Peer review of "S1. Data materials"

_Biogeosciences, 2019_

## Referee Comment (RC1) · Ben Bond-Lamberty (Referee) · 25 Aug 2019

General comments

————————-

This manuscript describes a review and synthesis of greenhouse gas (GHG) emissions from drained organic soils. This is an interesting and important topic, at least regionally, and appropriate for Biogeosciences. The text is well written, if a bit dense at times. I applaud the authors' goal of providing recommendations going forward.

[Figure]

There are a few problems. A few points in the text should be reconsidered for clarity or balance (see below). I was a bit surprised that neither the introduction nor the methods mentioned the global soil respiration database–see Bond-Lamberty and Thomson 2010 and https://github.com/bpbond/srdb–which seems relevant (but perhaps not?). Finally, one weakness of this kind of bespoke review is that it's not really reproducible, although the authors do a good job of describing their (somewhat subjective) criteria for inclusion/exclusion in the methods. Nothing really to do be done about this, but perhaps note it.

Overall, this is a careful and interesting synthesis that need minor revisions.

Specific comments

———————————-

1. Lines 102-104: might move this sentence somewhere more prominent, e.g. at very end of introduction

2. L. 225- and supplementary material: this seems a bit unbalanced to me. EC has strengths, such as integrating over a large spatial area, but it also has weaknesses–vulnerable to storage errors, low-turbulence conditions, advection, etc. See Wang et al. (2018, http://dx.doi.org/10.1016/j.agrformet.2017.07.023) and/or Barba et al. (2018, https://doi.org/10.1016/j.agrformet.2017.10.028) for example

3. L. 240-: confusing. Why are models the only way to quantify Rh? Later you mention trenched plots for instance

4. L. 284: probably start a new paragraph here for readability

5. L. 385-: agreed!

6. L. 468-: these are all good recommendations; what here is new/unexpected? That might be worth highlighting

7. Figure 1: this seems to omit CH4 from plants; is that intentional? Cf. Covey and

Megonigal (2018, 10.1111/nph.15624)

---

## Referee Comment (RC2) · Kees Jan van Groenigen (Referee) · 25 Sep 2019

I would like to echo the comments made by reviewer 1: this is a useful, detailed review on the current state of GHG flux estimates from drained organic forest soils. The authors discuss the limitations of several methods, knowledge gaps, and the data required to make GHG flux estimates suitable for extrapolation and data-synthesis efforts. My main concern is that parts of the manuscript are quite difficult to read. That is, the text is fairly dense, many sentences and some paragraphs are overly long. Starting at paragraph 5.4, the readability of the manuscript drastically improves. I encourage the

authors to adopt a similar writing style for the rest of the manuscript; it will probably reach a wider audience that way.

Some minor editorial suggestions:

L58: insert a comma after "Below the WT" L157: This title is not very clear. "Applicable" to what? To my mind, this section of the manuscript also doesn't really present a framework, as is suggested in the title. The first paragraph of the section is about challenges in CO2 flux measurements, the second paragraph is a comparison of different methods to estimate CO2 fluxes, the third paragraph is about the role of vegetation in determining CH4 and N2O fluxes etc. etc. L261: I wonder if this is a typo. What is "temperate region a"? L284: "Modelling" would be a good place to start a new paragraph. L308: "combine typically" = "typically combine" L333: Start a new sentence after "the estimates" L341: You could start a new paragraph after "In Tier 1.." L349: "...N2O emissions from ditches at...." L365: "carbon" = "C" L452: I assume you wrote "at least" because this statement might not be true for N2O and CH4, but this is not entirely clear.

---

## Author Comment (AC1) · 15 Oct 2019

AR: We thank the reviewer for the constructive comments on our manuscript. Here we give our response to these comments, and provide a modified manuscript.

RC: This manuscript describes a review and synthesis of greenhouse gas (GHG) emissions from drained organic soils. This is an interesting and important topic, at least regionally, and appropriate for Biogeosciences. The text is well written, if a bit dense at times. I applaud the authors' goal of providing recommendations going forward.

[Figure]

There are a few problems. A few points in the text should be reconsidered for clarity or balance (see below). AR: Thank you. These have all been considered as explained below.

RC: I was a bit surprised that neither the introduction nor the methods mentioned the global soil respiration database–see Bond-Lamberty and Thomson 2010 and https://github.com/bpbond/srdb–which seems relevant (but perhaps not?). Finally, one weakness of this kind of bespoke review is that it's not really reproducible, although the authors do a good job of describing their (somewhat subjective) criteria for inclusion/exclusion in the methods. Nothing really to do be done about this, but perhaps note it. Overall, this is a careful and interesting synthesis that need minor revisions. AR: Following this comment, we can easily agree that we could indeed mention the database somewhere in the paper. Reference to this database is added in the methods section (L 130-133). After all, there is also partial overlap with our database. Why it was not done in the first place is because our review is mostly aiming at providing guidance for future measurers concerning how to make their data more useful (for efforts such as the global soil respiration database). To keep the review within some limits, we have had to leave out a lot of material and references that are really important in this context in general, but seemed not to directly serve this overall aim. Drawing those lines has not been easy. We started with clearly more material than what is included currently. The global database seemed to be mostly useful for those who are already a few steps ahead and with the main in compiling data syntheses, which is not the focus

Specific comments: RC: 1. Lines 102-104: might move this sentence somewhere more prominent, e.g. at very end of introduction AR: We appreciated the suggestion, and tentatively moved the sentence to a later location. However, that appeared to cause some structural issues in both this and the following paragraph. This paragraph would end rather abruptly without the statement, and a new ending statement should be written. The following paragraph presents the specific aims, and we felt that the "overall objective" statement is needed before those, rather than after. Instead of dealing with

these issues, we returned the statement to its original location, and hope that it may be acceptable.

RC: 2. L. 225- and supplementary material: this seems a bit unbalanced to me. EC has strengths, such as integrating over a large spatial area, but it also has weaknesses– vulnerable to storage errors, low-turbulence conditions, advection, etc. See Wang et al. (2018, http://dx.doi.org/10.1016/j.agrformet.2017.07.023) and/or Barba et al. (2018,https://doi.org/10.1016/j.agrformet.2017.10.028) for example AR: The comment hits the point - every method have potential weaknesses that can emerge usually due to technical reasons in the method itself or due to conditions at the sites. Unfortunately number of Eddy covariance studies conducted in the sites covered by this review are very limited (3), which somewhat restricts analyzing the method in these specific conditions. Challenges in field data collection do exist as provided in the examples provided by the reviewer. We appreciate the publications listed and include more content on potential weak points in the method (L150-151, L 242-244, and S1 L 138-141).

RC: 3. L. 240-: confusing. Why are models the only way to quantify Rh? Later you mention trenched plots for instance AR: We agree that the statement here was simply confusing. We deleted the statement here, since the issue is actually dealt with in more detail later in the text as well as in the Supplementary material.

RC: 4. L. 284: probably start a new paragraph here for readability AR: Done as suggested.

RC: 5. L. 385-: agreed! AR: ïĄŁ

RC: 6. L. 468-: these are all good recommendations; what here is new/unexpected? That might be worth highlighting AR: What is most novel in this review, is probably the systematic analysis of the extent of background/environmental data reported along with the GHG fluxes, and the plea with recommendations for what to measure and present additionally. All of the recommendations should appear self-evident to an experienced researcher who has worked on data syntheses involving forests, such as the reviewers,

who may actually not be considered as the main target audience of the paper. The most important points are the first two general ones, which we hope that all future measurers will read. :-) The rest represent more specific data needs, and have been arranged by general topic areas, but could probably be arranged in several ways.

RC: 7. Figure 1: this seems to omit CH4 from plants; is that intentional? Cf. Covey and Megonigal (2018, 10.1111/nph.15624) AR: CH4 emissions from plants are discussed in several locations in the text and we recommend including vegetation in the flux monitoring. IPCC (2014) recommends reporting vegetation presence in flux monitoring, but does not obligate vegetation presence (see also L192). Therefore this detail was left out from the graph.

Please also note the supplement to this comment:
https://www.biogeosciences-discuss.net/bg-2019-261/bg-2019-261-AC1-supplement.pdf

─────────────────────────────

**Supplement:**

**Reviews and syntheses: Greenhouse gas exchange data from** drained organic forest soils – a review of current approaches and recommendations for future research**

- Jyrki Jauhiainen1,2, Jukka Alm3, Brynhildur Bjarnadottir4, Ingeborg Callesen5, Jesper R. Christiansen5, Nicholas Clarke6, Lise Dalsgaard7, Hongxing He8, Sabine Jordan9, Vaiva Kazanavičiūtė10, Leif Klemedtsson11, Ari Lauren3, Andis Lazdins12, Aleksi Lehtonen1, Annalea Lohila13,14, Ainars Lupikis12, Ülo Mander15, Kari Minkkinen2, Åsa Kasimir11, Mats Olsson9, Paavo Ojanen2, Hlynur Óskarsson16, -Bjarni D. Sigurdsson16, Gunnhild Søgaard7, Kaido Soosaar15, Lars Vesterdal5, and Raija Laiho1 5
- 10

1Natural Resources Institute Finland (Luke), Box 2, FI-00791 Helsinki, Finland

[revised manuscript text omitted]

(1) Number (and proportion) of papers included in the database that provide the specified information
 (2) For planted sites time of planting given with precision from year to a decade
 (3) Specific or average values for the monitoring site soil characteristics, not minimum/maximum or a range
 (4) Site type based on defined generally applied classification system
 (5) Not countable from the papers in unambiguous way for comparisons

(6) Not countable from the papers in unambiguous way as the data collection periods were described, for example, as 'snow-free period', 'warm season', or as a period between two dates

Figure 1: CO2, CH4 and N2O fluxes and mass transfer components contributing to soil C-stock changes in a forest ecosystem on drained organic soil, as in IPCC (2014). Arrows indicate flux/transfer direction.

---

## Author Comment (AC2) · 15 Oct 2019

AR: We thank the reviewer for the constructive comments on our manuscript. Here we give our response to these comments, and provide a modified manuscript.

RC: I would like to echo the comments made by reviewer 1: this is a useful, detailed review on the current state of GHG flux estimates from drained organic forest soils. The authors discuss the limitations of several methods, knowledge gaps, and the data required to make GHG flux estimates suitable for extrapolation and data-synthesis efforts. My main concern is that parts of the manuscript are quite difficult to read. That is, the text is fairly dense, many sentences and some paragraphs are overly long. Starting at paragraph 5.4, the readability of the manuscript drastically improves. I encourage the authors to adopt a similar writing style for the rest of the manuscript; it will probably reach a wider audience that way. AR: Following this comment, we have gone through the text and tried to make it a bit easier for the reader. These actions included dividing very long sentences into shorter ones, deleting repetitive phrases, and introducing new paragraph breaks. All changes in the text that are not linked to a specific comment represent this revision. We kept it relatively modest, however, so as not to change the actual contents of the paper. We hope that it has been at least somewhat sufficient to make the text more reader-friendly, also bearing in mind that the reviewers' task is much harder than that of the "regular reader".

Some minor editorial suggestions: RC: L58: insert a comma after "Below the WT" AR: Done as suggested.

RC: L157: This title is not very clear. "Applicable" to what? To my mind, this section of the manuscript also doesn't really present a framework, as is suggested in the title. The first paragraph of the section is about challenges in CO2 flux measurements, the second paragraph is a comparison of different methods to estimate CO2 fluxes, the third paragraph is about the role of vegetation in determining CH4 and N2O fluxes etc. etc. AR: Fair Observation. Title is revised to outlined so that it mirrors the text content correctly (L 216).

RC: L261: I wonder if this is a typo. What is "temperate region a"? AR: Typo. Corrected (L277).

RC: L284: "Modelling" would be a good place to start a new paragraph. AR: Done as suggested.

RC: L308: "combine typically" = "typically combine" AR: Done as suggested.

RC: L333: Start a new sentence after "the estimates" AR: Long sentence is split into two (L359).

RC: L341: You could start a new paragraph after "In Tier 1.." AR: Done as suggested.

RC: L349: "...N2O emissions from ditches at...." AR: Missing preposition added.

RC: L365: "carbon" = "C" AR: Done as suggested.

RC: L452: I assume you wrote "at least" because this statement might not be true for N2O and CH4, but this is not entirely clear. AR: More specific expression applied (L483).

Please also note the supplement to this comment:
https://www.biogeosciences-discuss.net/bg-2019-261/bg-2019-261-AC2-supplement.pdf